# Osteopathic Manipulative Treatment Decreases Hospital Stay and Healthcare Cost in the Neonatal Intensive Care Unit

**DOI:** 10.3390/medicines9100049

**Published:** 2022-09-21

**Authors:** Hannah Roland, Amanda Brown, Amy Rousselot, Natalie Freeman, J. Michael Wieting, Stephen Bergman, Debasis Mondal

**Affiliations:** 1DeBusk College of Osteopathic Medicine, Lincoln Memorial University, 9737 Cogdill Road, Knoxville, TN 37932, USA; 2DeBusk College of Osteopathic Medicine, Lincoln Memorial University, 6965 Cumberland Gap Parkway, Harrogate, TN 37752, USA

**Keywords:** neonatal intensive care unit, NICU, osteopathic medicine, osteopathic manipulative treatment, hospital stay, healthcare cost

## Abstract

Osteopathic manipulative treatment (OMT) is used in both inpatient and outpatient settings. Evidence suggests that OMT can reduce both patients’ recovery time and the financial cost of their acute medical treatment and rehabilitation. Multiple studies from neonatal intensive care units (NICUs) are presented in this article that demonstrate infants treated with OMT recover faster, are discharged earlier, and have lower healthcare costs than their non-OMT-treated counterparts. Data clearly show that adjunctive OMT facilitates feeding coordination in newborns, such as latching, suckling, swallowing, and breathing, and increases long-term weight gain and maintenance, which reduces hospital length of stay (LOS). Osteopathic techniques, such as soft tissue manipulation, balanced ligamentous tension, myofascial release, and osteopathic cranial manipulation (OCM), can reduce regurgitation, vomiting, milky bilious, or bloody discharge and decrease the need for constipation treatment. OMT can also be effective in reducing the complications of pneumonia in premature babies. Studies show the use of OCM and lymphatic pump technique (LPT) reduces the occurrence of both aspiration and environmentally acquired pneumonia, resulting in significantly lower morbidity and mortality in infants. Based on published findings, it is determined that OMT is clinically effective, cost efficient, a less invasive alternative to surgery, and a less toxic choice to pharmacologic drugs. Therefore, routine incorporation of OMT in the NICU can be of great benefit in infants with multiple disorders. Future OMT research should aim to initiate clinical trial designs that include randomized controlled trials with larger cohorts of infants admitted to the NICU. Furthermore, a streamlined and concerted effort to elucidate the underlying molecular mechanisms associated with the beneficial effects of OMT will aid in understanding the significant value of incorporating OMT into optimal patient care.

## 1. Osteopathic Manipulative Treatment: A Brief Overview

Osteopathic manipulative medicine (OMM) is an evidenced-based comprehensive approach to healthcare in which osteopathic physicians apply osteopathic philosophy, structural assessment, and the use of osteopathic manipulative treatment (OMT) in the diagnosis and treatment of patients by emphasizing that structure and function are integrated in health and disease states. Indeed, the science and art of OMM is becoming increasingly popular in the healthcare field, primarily due to its clinical effectiveness and minimally invasive nature [1,2,3]. Osteopathic manipulations consist of a set of patient-focused hands-on treatments directed at adjusting the body’s musculoskeletal system. The osteopathic method of patient care was designed to be a “complete system”, which means that OMT can be directed and modified for any disease type and severity. Most importantly, the osteopathic philosophy with a holistic approach is based on health enhancement and preventive care that emphasizes the integration of structure and function to treat the mind, body, and spirit of the patient and not just temporary relief of symptoms. Therefore, the four tenets of osteopathic medicine are defined as follows: (i) the body is a unit, and the person is a unity of body, mind, and spirit; (ii) the body is capable of self-regulation, self-healing, and health maintenance; (iii) structure and function are reciprocally interrelated; and (iv) a rational treatment is based upon an understanding of the basic principles of body unity, self-regulation, and the interrelationship of structure and function. As a result, OMT is now being considered as both a standalone treatment and for use as an adjunct to conventional interventions [4]. At this juncture, it is important to ensure readers understand the fundamental difference between OMT/manipulation (the therapeutic application of manually guided forces by an osteopathic physician to improve physiologic function and/or support homeostasis that has been altered by somatic dysfunction) as done by doctors of osteopathic medicine (DO) and others, and osteopathic medicine, which is defined as a complete system of medical care practiced by DOs with an unlimited license to practice the full scope of medicine, that is represented by a philosophy that combines the needs of the patient with the current practice of medicine, surgery, etc., where the use of hands-on patient management process is incorporated with the goal of achieving maximum painless movement of the musculoskeletal system. 

The use of OMT in neonatal intensive care units (NICUs) has demonstrated tangible successes in the treatment of pneumonia, ankyloglossia, feeding dysfunction, and gastrointestinal distress in preterm babies as well as in assisting with weight gain and maintenance of newborns after discharge. A recent prospective multicenter observational study (OSTINF Study) found that osteopathic treatment is associated with major positive changes in the severity of health complaints in neonates and infants [5]. Therefore, a growing number of parents are turning to osteopathic physicians for help with their infants’ and children’s chronic health issues. 

The primary basis of osteopathic medicine is the belief that somatic dysfunctions can result in associated conditions impacting peripheral nerves, blood flow, immune system efficacy, and organ function. In general, the goal of osteopathic treatment is to realign the body and motivate self-healing at both a superficial and deeper level by manipulating the fascia [6,7]. Fascia is a large fabric of connective tissue surrounding all muscles, ligaments, vasculature, nerve fibers, organs, and bones that is made up of collagen, elastin, and a liquid component, the ground substance. Fascia gives muscles shape, elasticity, flexibility and, most importantly, lubrication. When fascia is stretched or compromised, it becomes tighter. Several OMT techniques are focused on lengthening and correcting the myofascia to allow corresponding muscles to return to a meaningful, pain-free state, which can be achieved by applying consistent pressure and stretch on the fascial network to aid tissue release and achieve homeostasis. A balance in the fascial system increases the ease of motion, facilitates relaxation, and decreases pain. In this respect, it is increasingly clear that our bodies are very adaptive, especially the fascial tissue. 

Furthermore, because fascia is part of the interstitial space, which is a primary source of lymph and a major fluid compartment in the body, it is most likely that the effects of osteopathic manipulations on the fascia also affect the fluid dynamics at the applied tissue sites and reduce the edematous condition and inflammatory activity [8]. In addition, because the fascia also contains innervations of the autonomic nervous system (ANS), both parasympathetic and sympathetic nerves, which play a decisive role in regulating neurogenic inflammation at different tissue locations following injury [9], the application of OMT may also have positive effects in regulating normal functioning of the nervous system [10]. Indeed, studies in rats have clearly documented the influence of sympathetic nervous system on capsaicin-evoked enhancement of dorsal root reflexes [11]. The neural basis of the osteopathic lesion was first reported by Korr et al. as early as 1947 [12], and Denslow et al. (1975) further corroborated the pathophysiologic significance of inflammation within osteopathic lesion [13]. Therefore, the multimodal actions of OMT on the fascia, musculature, lymph, blood flow, and the nervous system, could be the basis of improvements in clinical conditions observed in patients treated with different OMTs. The different OMT techniques include, but are not limited to, myofascial release, balanced ligamentous tension, lymphatic pumping, rib raising, diaphragmatic manipulation, high-velocity low-amplitude manipulation, and osteopathic cranial manipulation. These are specific techniques that DOs are trained to perform. Furthermore, several osteopathic techniques can be modified based on the patient’s condition, age, weight, and other characteristics or underlying diseases, yielding a personalized approach with lower side effect potential. 

In the following sections, a brief overview of common OMT techniques used in the NICU setting are provided, followed by clinical findings of their benefits from around the globe. These findings demonstrate the effectiveness of OMT in treating neonates and infants and provide evidence that adjunctive OMT can significantly decrease the cost of medical care and the length of hospital stay (LOS). The following five OMT techniques are demonstrated to be safe and effective in neonates and infants admitted to the NICU: (A) lymphatic pump technique (LPT); (B) diaphragm treatment (DPT); (C) balanced ligamentous tension (BLT); (D) myofascial release (MFR); and (E) osteopathic cranial manipulation (OCM) (Figure 1).

### 1.1. Lymphatic Pump Technique

The lymphatic system, which is essential for fluid balance, fatty acid absorption, and immune system regulation, plays a crucial role in collecting and directing tissue fluid, immune cells, antigens, pathogens, and proteins from tissue interstitial space and delivering them to regional lymph nodes [14]. Internal movements, such as muscle contraction, intestinal motility, and respiratory motions, as well as external forces, such as exercise, limb movements, and OMT, can facilitate the flow of lymph through lymphatic vessels, ultimately resulting in significant health benefits [14,15]. In addition to fighting microbial infections, LPT has often been used by osteopathic physicians for management of congestive heart failure and its effects, upper and lower gastrointestinal dysfunctions, respiratory diseases, and edema [2]. These aid in the circulation of lymphatic fluids and facilitate wound healing by increasing blood circulation. The efficacy of LPT in boosting the immune system has also been corroborated in several animal models [16]. Osteopathic physicians frequently use a combination of LPTs for different regions of the body to achieve optimal lymphatic circulation. Therefore, the utility of LPT in the NICU has been a major focus of osteopathic physicians [17].

### 1.2. Diaphragm Treatment

Diaphragm, the main inspiratory muscle, generates craniocaudal movement of its dome during contraction and relaxation [18]. The thoracic diaphragm is a domed circular muscle that is essential for proper respiratory functions, and it relaxes and contracts with each breath. Pathological changes may affect the diaphragm’s ability to raise and expand the lower rib cage, which leads to a decrease in the transverse diameter of the lower ribcage during inspiration. Therefore, both diaphragmatic stretch (DS) and manual diaphragm release (MDR) techniques, which are targeted at freeing tension in a specific region of the diaphragm, can be utilized [19]. Because the fascia surrounds and holds every organ, blood vessel, bones, nerve fiber, and muscle in place, including the attachments within and around the diaphragm, diaphragmatic dysfunctions and related biomechanical problems can arise due to tension within the fascia and may affect various bodily functions and musculoskeletal regions. Furthermore, because the diaphragm is crucial to respiration, restrictions can cause difficulty in breathing, which is especially evident in newborns [20,21]. Different diaphragm corrective treatments using osteopathic manipulation are being considered in babies with asthma, pneumonia, and other respiratory diseases [18,19,20].

### 1.3. Soft Tissue Technique

This is a category of treatments involving steady pressure, stroking, stretching, kneading, and separation of tissue origin and insertion, which is achieved by targeting the myofascia using different manipulative treatments. Indeed, a clear identification of myofascial dysfunction requires complete patient history and physical evaluation based on the presenting complaint. Studies show that a series of osteopathic treatments, whether soft tissue, myofascial release, or cranial techniques, can significantly improve motor function in children with moderate to severe spastic cerebral palsy [22]. Therefore, high-quality research to study the comparative effectiveness of these widely utilized noninvasive and easily applicable forms of OMT is clearly needed.

### 1.4. Balanced Ligamentous Tension

Ligaments connect bone to bone and, as such, span the joints of the body. Balanced ligamentous tension (BLT) is an indirect and passive technique that targets strain within ligamentous structures. BLT can target joints of the pelvis, spine, shoulders, knees, elbows, etc. In general, BLT technique uses the principle that disengagement of the diagnosed somatic dysfunctions can relieve underlying pain and morbidity [23]. In addition, some studies have also utilized balanced membranous tension (BMT), which targets the movement of intraosseous and dural membranes to a precise point, allowing the body to equalize the stress in all directions. When performing BLT, the osteopathic physician locates the site of restriction and carries the body to a balance point and then waits for a change in the palpatory quality of the structure being treated, i.e., a change in skin tension, temperature, or muscle tension [24]. At this point in the technique, the body is slowly correcting the diagnosed restriction toward healing. 

### 1.5. Myofascial Release

Myofascial release (MFR) can be performed both as a direct technique, by engaging restrictive barriers, and as an indirect technique, by moving away from barriers, with balanced tension. The MFR technique manipulates the myofascial complex to release binding tissues that are impeding normal muscle function. This will help stimulate the body’s healing processes, which facilitates improvements in restriction, restores function, and decreases patient discomfort, all of which improves overall health [25]. Although there is limited research in the effectiveness of BLT and MFR in neonates and infants, the available clinical findings show improved motor function in children with moderate to severe spastic cerebral palsy [22,26]. 

### 1.6. Osteopathic Cranial Manipulation

In addition to dysfunctions within the peripheral nervous system, osteopathic physicians are well trained to provide balance to the central nervous system (CNS) through osteopathic cranial manipulation (OCM). The benefits are clearly evident in neonates, especially those manifesting plagiocephaly, or misshapen head, which results from abnormal distortion of the cranial bones that can occur in utero, during birth, or afterward. Plagiocephaly and feeding difficulty are two of the most commonly treated presentations in the NICU. Indeed, one of the most important uses of OCM is to correct plagiocephaly [27,28]. Craniosynostosis, on the other hand, may warrant surgical intervention as craniosynostosis is a more severe form of misshapen head resulting from premature fusion of a cranial suture. Craniosynostosis may result in developmental craniofacial anomalies and in the impairment of brain development in addition to an abnormally shaped skull [29]. 

The reliability of diagnostic procedures for early brain dysfunction and the efficacy of OCM in correcting them in the long term has also been shown recently [30]. OCM can significantly improve somatic dysfunctions and parasympathetic regulations [23]. Although surgical techniques have been most often employed to correct craniosynostosis in neonates, several positional holds of the cranium have been shown to affect parasympathetic output from the vagus nerve and thus impact other processes that cause somatic dysfunctions. Cranial manipulations help balance the flow of cerebrospinal fluid (CSF) and maintain the inherent motility of the brain and may also be involved in regulating respiratory mechanisms via the CNS.

## 2. Evidence-Based Medicine: OMT Is Effectively Implemented in the NICU

Following delivery, several factors contribute to the need for extended length of stay (LOS) in the NICU [31,32]. Historically, the population requiring most NICU admissions is preterm births, defined as a delivery prior to 37 weeks of gestation. Approximately 10% of all births require admission to the NICU, secondary to premature birth [33]. This results in a disproportionate amount of overall NICU cost that is associated with preterm care. A major concern of both term and preterm infants in the NICU is difficulty with breastfeeding and maintenance of the infant’s growth after discharge [34,35]. Furthermore, in the current era of increasing nosocomial infection, both term and preterm infants admitted to the NICU are particularly vulnerable to respiratory and gastrointestinal infections [36,37,38]. Therefore, there is a need for alternative, safe treatment options that have demonstrated promise in significantly reducing the need for hospitalization and LOS in the NICU. In this respect, OMT has often been found to be effective in combating prolonged hospital stays in infants [39,40,41,42,43]. In the following sections, we present findings from several NICU clinics around the world that show the significant efficacy of OMT in reducing infant morbidity and mortality (Figure 2). 

### 2.1. OMT as a Preventive Measure against Pneumonia in Infants

Infants and elderly are at the highest risk of respiratory tract infections and disease progression toward lung consolidation and pneumonia. Indeed, respiratory infections are the leading infectious cause of hospitalization and death among elderly and a leading cause of hospitalization in children [44]. This includes pneumonia not only caused by complications of aspiration but also by environmental exposure. Although immunizations against common pathogens associated with respiratory infections are effective in preventing pneumonia progression, the pre-vaccination status in infants and the waning vaccine efficacy in elderly makes them both susceptible to pathologic progression [45]. Pneumonia accounts for 15% of childhood mortality under the age of five and is most common amongst children with poor nutrition and weakened immune systems [46]. Pediatric aspiration pneumonia has also been linked to higher rates of hospital admissions and readmissions, longer lengths of stay, and increased median cost of hospitalization [47]. Furthermore, the frequency of aspiration pneumonia in infants is found to increase as the gestational age in mothers decreases [48]. This is of particular interest as the pregnancy rates for women aged 35–39 and 40–45 years has increased by 47% and 80%, respectively [49]. Physicians often prescribe broad-spectrum antibiotics to prevent this progression from mild respiratory infections to overt pneumonia. However, antibiotic overuse has been linked to resistance development and increased adverse effects [50]. Therefore, availability of safe and effective approaches to prevent pneumonia in children, which results in increased NICU admission, would be an essential measure to reduce childhood mortality [51]. 

In this respect, the application of OMT can be of significant benefit for the prevention of both hospital-acquired (nosocomial) and community-acquired pneumonia (CAP) [52]. Adjunctive OMT has been shown to aid recovery from pneumonia by both enhancing the functions of the immune system and maximizing the effects of antibiotics [53]. The utility of osteopathic medicine, which accentuates the body’s innate principles of self-healing and achieves homeostasis, will thus be of significant translational value. Furthermore, because the treatment plan and the quantity of OMT sessions vary according to the patient and course of the disease, OMT may be a better alternative to antibiotics in the NICU. Indeed, past studies have demonstrated the benefits of OMT in suppressing inflammation and disease progression in patients infected with the influenza virus [54,55], and the benefits of OMT had been documented during the Spanish flu pandemic in 1917 [56]. Due to its efficacy and noninvasive nature, several clinical trials are underway to investigate whether adjunctive OMT can decrease disease progression in hospitalized COVID-19 patients as well [57,58]. Therefore, a thorough evaluation of the benefits of integrating OMT in both neonates and infants admitted to the NICU with respiratory infections is clearly needed.

### 2.2. Potential of OMT in Decreasing Ankyloglossia in Newborns

Both ankyloglossia and cranial nerve dysfunctions, and the associated restrictions in tongue movements, are common in pediatric patients. Ankyloglossia is the result of an unusually short frenulum (the tissue that anchors the tongue to the floor of the oral cavity) or accumulation of excessive tissue in the same region [59,60]. The greatest concern amongst newborns with this condition is its implications on feeding. Tobey et al. (2018) [61] reported 17 children with diagnosis of ankyloglossia and found 74.6% had difficulty latching on to the nipple and 35% had difficulty sucking. It was also reported that a large percentage (94%) of participating neonates had dysfunctions or restrictions of at least one cranial suture that often involved a region between the skull and the first vertebra (atlantooccipital joint). Most notably, all infants had findings of occipital condylar dysfunction, defined as two or more of the following: localized tenderness, musculoskeletal asymmetry, restriction of motion, or tissue texture abnormalities [61]. In this respect, a variety of osteopathic techniques have demonstrated the ability to correct somatic dysfunctions at the cranial base, and hence it is postulated that ankyloglossia in newborns may be ameliorated by early exposure to OMT [62]. Hawk et al. (2018) published a comprehensive review that showed numerous reports on the potential of OMT to correct cranial nerve dysfunctions and decrease ankyloglossia in newborns [63]. Furthermore, Herzhaft-Le Roy et al. (2017), demonstrated that incorporating osteopathic treatment into regular lactation consultations is both beneficial and safe in correcting infants with biomechanical sucking difficulties [64]. These findings clearly demonstrate that OMT has the potential to reduce ankyloglossia and cranial nerve dysfunctions in newborns.

### 2.3. OMT Facilitates Feeding Coordination in Newborns

The ability of infants to regulate their temperature, stabilize breathing, and gain weight is the most common criterion for NICU discharge [65]. Consequently, one major cause of delay in discharge is their inability to feed. In this respect, the beneficial effect of adjunctive OMT in decreasing nipple feeding dysfunction has been shown on several occasions [66,67,68]. A case study from Lund et al. (2011) involved the use of OMT in the treatment of two premature twins whose inability to feed prompted discussions about feeding tube placement [40]. During the study, the type of OMT method used was determined by exam findings unique to each infant, and techniques such as soft tissue manipulation, BLT, MFR, and/or OCM were utilized. The duration of treatment and technique used were left to the discretion of the treating osteopathic physician. Findings showed that the twins’ ability to nipple feed gradually improved after OMT was initiated. After three weeks of treatment, both infants were nipple feeding without complications due to improved oral coordination required for nipple feeding. 

One of the most significant factors that prolongs LOS is delayed transition from gavage to nipple feeding, which has been shown to decrease in premature infants who received OMT [66,67]. Cerritelli et al. (2014) defined key steps for a rigorous and effective osteopathic approach that can be safely and effectively used in neonates to decrease their hospital stay [66]. In 70 premature infants (control group = 35 and osteopathic group = 35), a 5-day reduction in LOS (*p* = 0.042) was documented in the osteopathic group compared to the control group, with a greater effect observed in infants with very low birth weight [67]. Conversely, a more recent study by Danielo Jouhier et al. (2021), which included 128 mother–infant dyads, showed that OMT did not significantly improve feeding coordination and exclusive breast feeding at 1 month [68]. Therefore, while OMT is relatively safe in most cases, the above contradictory findings suggest that additional well-constructed clinical trials should be conducted to verify the therapeutic benefits of OMT in the NICU.

### 2.4. OMT Decreases Gastrointestinal Dysfunctions and Assists in Weight Gain

Stress-induced damage to the gastric mucosa occurs frequently in neonates [69]. Numerous studies have documented that OMT is an effective strategy for reducing gastrointestinal (GI) dysfunction in preterm infants [41,42]. Pizzolorusso et al. (2011) showed that infants who received 20 min biweekly sessions of OMT within 14 days of birth experienced 55% less regurgitation, vomiting, and milky bilious/bloody discharge (from infants with oro/nasogastric tubes) and significantly decreased the need for treatment of constipation [41]. Consequently, infants receiving OMT were able to be discharged significantly earlier than their control counterparts, thus reducing overall LOS by ~75% [41]. Limitations of this study included a small population size (study group *n* = 162) and limited demographic information, which may have an impact on overall external validity. 

In another multicenter, randomized, controlled trial, newborns were assigned to either standard prenatal care (control group) or OMT (treatment group), and the effects on LOS were documented [42]. The primary outcome was the mean difference in LOS between groups, with the OMT group showing a reduction of 3.9 days from a mean of −5.5 days to −2.3 days (95% CI, *p* < 0.001). Cicchitti et al. (2020) [70] also demonstrated that OMT may represent an effective support to the standard of care in newborns admitted to the NICU, underscoring the idea that incorporation of osteopathic treatment as a supplement to standard practice would be of great interest [66,68,70].

## 3. OMT Reduces the Cost of Medical Care in the NICU

### 3.1. Increasing Cost of Medical Care in the NICU

In 2007, the economic burden of preterm infant care in the NICU was approximately $5.8 billion [71], accounting for 47% of all infantile care costs. This equates to a hospital bill of nearly $40,000 for preterm infants versus around $1000 for their term counterparts [72]. Unfortunately, NICU admission rates have also been steadily increasing, rising from 6.4% in 2007 to 7.8% in 2012 [31,72]. Two subsequent publications by Hamilton et al. (2019 and 2021) also predicted a continued rate of progression of NICU admission to as much as 10.23% [73,74]. 

### 3.2. OMT Reduces LOS and the Cost of Medical Care in the NICU

Published findings from NICU clinics around the world clearly support the fact that OMT has the potential to reduce overall cost of neonatal health services by shortening the need for infants to continue to stay in the NICU. Cumulative data from numerous studies demonstrate the ability of OMT to expedite NICU discharge and are discussed more in depth in the following sections, and the comparative data analysis from three of these publications is provided in a bar graph (Figure 3).

Substantive evidence for a positive impact of OMT in reducing the need for hospitalization comes from Cerritelli et al. (2015), which involved the inclusion of OMT into treatment plans in preterm infants admitted to the NICU [42]. This study included data from three different public hospitals and administration of OMT roughly 4 days after birth [42]. OMT treatments used in the study were similar to that in previous preterm studies, being either indirect MFR or BLT release, with each session lasting 30 min. In an effort to maintain internal validity, the study design blinded NICU staff and involved osteopathic physicians that did not participate in patient care decision-making. Resulting data indicated an average hospital stay reduction of 4 days (*p* < 0.001) for the OMT study group and an associated $1816.22 (€1586.01) (*p* < 0.001) cost reduction per infant. Altogether, OMT saved a total of $572,575.00 (€500,000) over the 14 months the study was active [42]. It is to be noted that this study involved a larger population, totaling 695 participants across three separate public NICUs. 

A recent study by Cicchitti et al. (2020), included another large cohort of NICU admissions and utilized retrospective data collected from 2008 to 2016 at the Santo Spirito Hospital in Pescara, Italy [70]. The study design included babies who received standard care in conjunction with OMT (*n* = 652) and a control group who received only standard care (*n* = 597). Nearly 49% of the babies were preterm infants (315 in the control group and 296 in the experimental group). The findings showed that the experimental group had undergone a marked weight gain of 83 g vs. 35 g for the control group (*p* < 0.001). The findings also showed that infants in the full-term group required less OMT sessions versus the preterm infant subgroup (two vs. three treatments, respectively). Further breakdown of the preterm subgroup showed that those born at 27–31 weeks of gestation required seven sessions on average in contrast to four sessions for babies born at 32–33 weeks. Additional study findings included a reduction in LOS between premature babies (defined as gestational age of 27–31 weeks) in the OMT group as compared to those in the control group. Overall, the preterm infants who received OMT had a mean LOS of only 27.5 days versus as much as 61.3 days in the non-OMT group (control). However, a similar trend was not observed in term infants [70]. 

Pizzolorusso et al. (2014) demonstrated that the application of two osteopathic treatments to moderate preterm and late preterm infants (32–36 weeks gestation) resulted in average savings of $777.11 (€740.00) per day per infant by reducing the duration of their hospitalization [75]. Additional data analysis from this study revealed that the LOS was significantly reduced when OMT was integrated into treatment plans early in admission. Infants who received early OMT (0–4 days after birth) had a 4-day average LOS reduction, whereas infants who received late OMT (0–14 days after birth) experienced a 2-day average LOS reduction [75]. In an earlier study in both moderate and preterm infants, Pizzolorusso et al. (2011) had also documented that optimally timed OMT can reduce hospital LOS by as much as 75% [41]. Data from this previous study clearly showed that infants who received biweekly OMT experienced 55% less GI symptoms following feeding, allowing them to be discharged significantly earlier than their control counterparts. 

Similar reductions in hospital stay have been observed by Cerritelli et al. (2013) who recruited eight osteopathic physicians to perform routine 20 min osteopathic examinations and treatments to NICU patients at Santo Spirito Hospital in Pescara, Italy [76]. Infants enrolled in the study were born at a gestational age of >28 and <38 weeks. A total of 110 randomized patients began the study, with 9 individuals being excused at varying times throughout the study due to being transferred to other facilities. In the end, the experimental and control groups included 47 and 54 infants, respectively. Results from the study demonstrated a mean LOS of 31.3 +/− 20.2 for the control group versus 26.1 +/− 16.4 for the experimental group (*p* < 0.3). This equated to 6 days of hospital stay reduction for preterm infants treated with OMT. With respect to cost, the experimental group had net savings of $3387.80 (€2958.39) per preterm per length of stay as their total cost was $5143.68 (€4491.71) for the OMT group vs. $8531.47 (€7450.09) for the control group. By the end of the 15-month study period, the experimental group had saved an estimated $159,226.58 (€139,044.30) [76]. 

It is noteworthy that there are significant limitations in the above studies, especially the small effect sizes and lack of heterogenous population. Furthermore, most of the afore-mentioned studies were performed within single hospital settings, often within the same country. Only a small number of these studies on OMT in the NICU used data from multiple hospitals or care centers. Therefore, future research should seek participation from a variety of institutions to enable larger population sizes and more diverse demographics. Such criterion-compliant research can also help substantiate the claim that OMT significantly benefits neonates and allows for shorter and more affordable hospital stays. In addition, the reader is encouraged to refer to another of our published article in this Special Issue of Medicines, which provides a more comprehensive review of various OMT approaches and their therapeutic uses [77].

## 4. Conclusions

This review article discusses data from multiple clinical studies that clearly demonstrate the therapeutic benefits of OMT in decreasing both recovery time and the financial cost of medical treatment and rehabilitation in the NICU. Studies show that adjunctive OMT helps feeding coordination in newborns and decreases the need for constipation treatment. Furthermore, OMT can be highly effective in reducing the complications of both aspiration and environmentally acquired pneumonia in premature babies. Five OMT techniques are most frequently used in the NICU setting, namely, lymphatic pump techniques, diaphragm treatment, balanced ligamentous tension, myofascial release, and osteopathic cranial manipulation, which have been discussed at length. 

The multi-institutional and randomized OSTINF Study found that OMT is associated with significant reductions in the severity of health complaints in neonates and infants [5]. However, despite encouraging findings, a concerted and streamlined effort to properly catalog the effectiveness of OMT in treating infants are still lacking. Furthermore, although osteopathic medicine is based upon an understanding of the basic principles of body unity, self-regulation, and the interrelationship of structure and function, the lack of investigations to delineate the underlying mechanisms associated with its therapeutic benefits remains a significant challenge. While several published studies have documented the financial benefits of using OMT in the NICU, as documented in this review, there is a dearth of information on how these “tried and true” osteopathic techniques achieve their outcomes at the molecular level. Elucidation of these mechanistic insights will not only enable us to understand the significant benefits of OMT treatment of newborns and infants but also help guide osteopathic treatment in adults and elderly populations. In this respect, Wieting et. al. (2013) showed that a daily postoperative OMT protocol improved functional recovery of patients who underwent a coronary artery bypass graft (CABG) [78]. Future OMT studies in older patients, especially those with chronic conditions, will both facilitate and underscore the efficacy of osteopathic manipulations.

Furthermore, development of both in vitro and in vivo models to study the beneficial effects of OMT and the myofascial changes occurring post-OMT would be of significant translational value [79]. Current understandings of the connections and cross-talks within the vascular, immune, and nervous systems and the systemic beneficial effects of a whole-body healing via OMT would need to be properly understood or, at the least, thoroughly discussed within the medical community. In addition, with the advent of new biomarker detection technologies, the effects of osteopathic treatment on changes in the patient’s microbiome [80] as well as changes in the micro-RNA (miRNA) profile within exosomes or extracellular vesicles (EVs) [81], especially those secreted in circulation post-OMT treatment, may provide highly valuable clues about their molecular mechanism/s and will be an essential guide for their optimal usage.

## Figures and Tables

**Figure 1 medicines-09-00049-f001:**
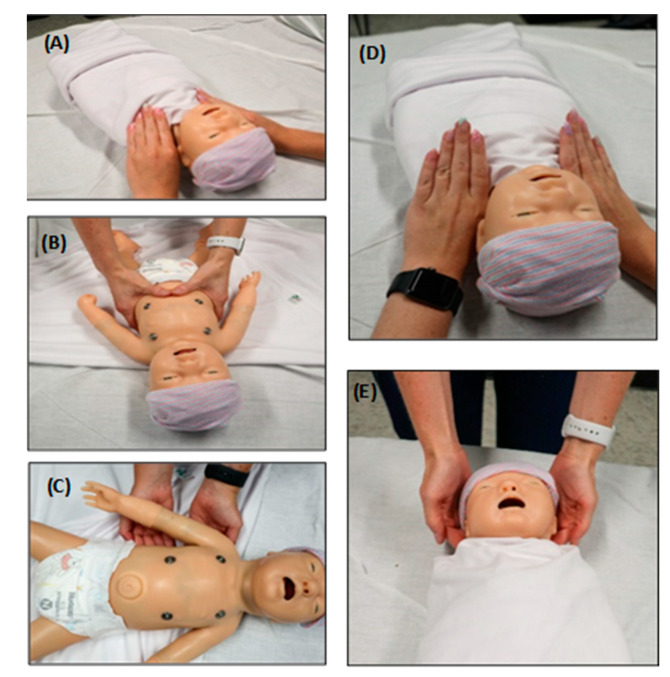
OMT techniques often utilized on babies in the NICU. The insets represent hand placement of different OMT techniques using a plastic neonate model. (**A**) Lymphatic pump technique (LPT). This technique uses a sequential pneumatic compressive pumping action that ameliorates swelling and increases immune surveillance by mimicking the lymphatic system. One such technique, the thoracic pump technique, is demonstrated in the image. (**B**) Diaphragm treatment (DPT). This technique can ameliorate a dysfunctional thoracic diaphragm and optimize its physiologic function. This technique also influences the thoracic and lumbar spine, ribs, and sternum. (**C**) Balanced ligamentous tension (BLT). This includes manipulation of joint ligaments to a precise balance point, allowing the body to minimize tension through an equalization of stress in all directions. Rib raising is one such technique shown in the figure. Fingers are placed on the rib angles, and an anterior pressure (toward the ceiling) is applied and held for varying times to balance the sympathetic chain. (**D**) Myofascial release (MFR). This is a broad category of OMT techniques directed at liberating the muscles and fascia from restrictions such as flexion, extension and side bending. Dysfunctional myofascial tissues can be loaded into the position of ease or into the “restrictive barrier” until the restriction is improved. A restrictive barrier is defined as a functional limit that abnormally diminishes the normal physiologic range of motion. The thoracic inlet technique is one example of MFR demonstrated in the image. (**E**) Osteopathic cranial manipulation (OCM). This primarily uses respiratory mechanisms and balanced membranous tension for cranial-related somatic dysfunctions. One highly utilized OCM technique in infants is the occipital release technique, which is shown in the image.

**Figure 2 medicines-09-00049-f002:**
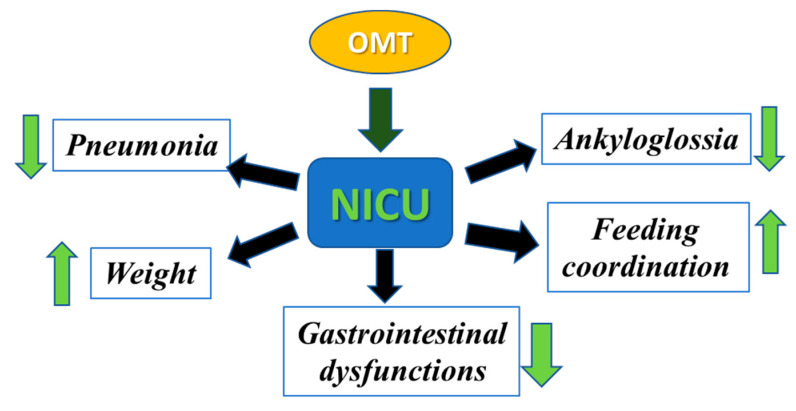
Efficacy of OMT in reducing morbidity and mortality in infants. Different osteopathic techniques can be used to decrease pneumonia, ankyloglossia, and gastrointestinal dysfunction and to increase feeding coordination and weight gain in infants admitted to the NICU.

**Figure 3 medicines-09-00049-f003:**
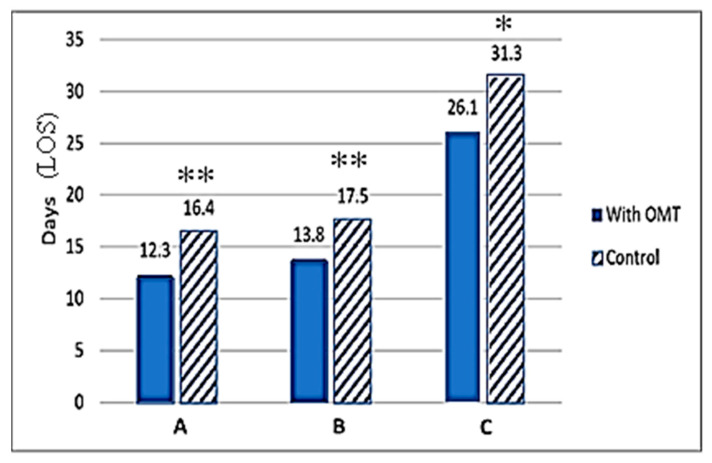
Effect of adjunct OMT on length of hospital stay (LOS). The number of days of NICU stay are presented on the X-axis. The hospital stays (days) are also presented at the top of each bar graph. Hatched bars represent control (no OMT), and solid bars show data obtained in the OMT group. Significant differences between control (non-OMT) and the experimental (with OMT) groups are designated by asterisk above the bars. Data represents published findings from (**A**) Pizzolorusso G. et al. (2014) [75] with ** denoting *p* < 0.001, (**B**) Cerritelli F. et al. (2015) [42] with ** denoting *p* < 0.001, and (**C**) Cerritelli F. et al. (2013) [76] with * denoting *p* < 0.03.

## Data Availability

The bar graph data included were generated from published findings from Pizzolorusso G. et al. (2014) [75], Cerritelli F. et al. (2015) [42] and Cerritelli F. et al. (2013) [76].

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
