# Peer review of "Osteopathic Manipulative Treatment Decreases Hospital Stay and Healthcare Cost in the Neonatal Intensive Care Unit"

_medicines, 2022, doi:10.3390/medicines9100049_

Round 1

Reviewer 1 Report

I consider worthwhile to extend the idea about the osteopathic treatment (70, 71) beyond the concept of manipulating the fascia.                                            The fascia, as described by  Theise ND (2018),

( Structure and Distribution of an Unrecognized Interstitium in Human Tissues.

Benias PC, Wells RG, Sackey-Aboagye B, Klavan H, Reidy J, Buonocore D, Miranda M, Kornacki S, Wayne M, Carr-Locke DL, Theise ND Sci Rep. 2018 Mar 27;8(1):4947. 

doi: 10.1038s41598-018-23062-6. PMID: 29588511 )

is part of the interstitial space, which is the primary source of lymph and is a major fluid compartment in the body. Then, arguably, the osteopathic treatment conditions the state of the fluids of the body, for instance reducing an edematous condition; in other words decreasing the inflammatory activity. The neurogenic inflammation and sympathetic activity are related to each other. It has been demonstrated that sympathetic efference plays a decisive role in the onset of inflammatory phenomena (Wang,J.; Ren, Y.; Zou, X.;Fang,L.;Willis,W.D.; Lin, Q. Sympathetic influence on capsaicin-evoked enhancement of dorsal root reflexes in rats. J. Neurophysiol. 2004, 92, 2017–2026) .
These findings agree with what Denslow and Korr underscored regarding Somatic Dysfunction, as it pertains to expressiveness of phenomena related to neurogenic inflammation  (Denslow, J. S. Pathophysiologic evidence for the osteopathic lesion: The known, unknown, and controversial.J.Am.Osteopath. Assoc. 1975, 75, 415–421)                                          and autonomic sympathetic innervation (Korr, I.M.The neural basis of the osteopathic lesion.J.Am.Osteopath.Assoc.1947,47,191–198).

This phenomenology could be at the basis of the improvement of the clinical conditions described in the article. 

Author Response

Please see the attached Cover Letter for answer to the reviewer's suggestion and comments.  Thank you.

Reviewer 2 Report

In this review article, Roland et al summarized the various OMT techniques used in the NICU, the benefits of OMT in newborns, and the cost-effectiveness of OMT implementation in the NICU. Overall, this review is thorough and well-written.

1. Line 226-240: the focus of this paragraph should be on the preventative role of OMT against development of pneumonia, and yet only the last sentence of this paragraph is devoted to this topic. Line 223-237 “This is of particular interest since…significantly higher odds of NICU admission” can be deleted. 

2. Section 4. Conclusions can be enhanced by summarizing the first three sections, in addition to discussing limitations and future directions. 

Author Response

(The authors gave the same response as above.)
